# The Vendi Score: A Diversity Evaluation Metric for Machine Learning

## Abstract

Diversity is an important criterion for many areas of machine learning (ML), including generative modeling and dataset curation. Yet little work has gone into understanding, formalizing, and measuring diversity in ML. In this paper we address the diversity evaluation problem by proposing the *Vendi Score*, which extends ideas from ecology to ML. The Vendi Score is defined as the exponential of the Shannon entropy of the eigenvalues of a similarity matrix. This matrix is induced by a user-defined similarity function applied to the sample to be evaluated for diversity. In taking a similarity function as input, the Vendi Score enables its user to specify any desired form of diversity. Importantly, unlike many existing metrics in ML, the Vendi Score does not require a reference dataset or distribution over samples or labels, it is therefore general and applicable to any generative model, decoding algorithm, and dataset from any domain where similarity can be defined. We showcase the Vendi Score on molecular generative modeling where we found it addresses shortcomings of the current diversity metric of choice in that domain. We also applied the Vendi Score to generative models of images and decoding algorithms of text where we found it confirms known results about diversity in those domains. Furthermore, we used the Vendi Score to measure mode collapse, a known shortcoming of generative adversarial networks (GANs). In particular, the Vendi Score revealed that even GANs that capture all the modes of a labelled dataset can be less diverse than the original dataset. Finally, the interpretability of the Vendi Score allowed us to diagnose several benchmark ML datasets for diversity, opening the door for diversity-informed data augmentation.[1]

## 1 Introduction

Diversity is a criterion that is sought after in many areas of machine learning (ML), from dataset curation and generative modeling to reinforcement learning, active learning, and decoding algorithms. A lack of diversity in datasets and models can hinder the usefulness of ML in many critical applications, e.g. scientific discovery. It is therefore important to be able to measure diversity.

Many diversity metrics have been proposed in ML, but these metrics are often domain-specific and limited in flexibility. These include metrics that define diversity in terms of a reference dataset (Heusel et al., 2017; Sajjadi et al., 2018), a pre-trained classifier (Salimans et al., 2016; Srivastava et al., 2017), or discrete features, like n-grams (Li et al., 2016). In this paper, we propose a general, reference-free approach that defines diversity in terms of a user-specified similarity function.

Our approach is based on work in ecology, where biological diversity has been defined as the exponential of the entropy of the distribution of species within a population (Hill, 1973; Jost, 2006; Leinster, 2021). This value can be interpreted as the effective number of species in the population. To adapt this approach to ML, we define the diversity of a collection of elements $x_1, \ldots, x_n$ as the exponential of the entropy of the eigenvalues of the $n \times n$ similarity matrix $K$, whose entries are equal to the similarity scores between each pair of elements. This entropy can be seen as the von Neumann entropy associated with $K$ (Bach, 2022), so we call our metric the *Vendi Score*, for the von Neumann diversity.

---

[1]Code for calculating the Vendi Score will be made available publicly after the anonymity period.

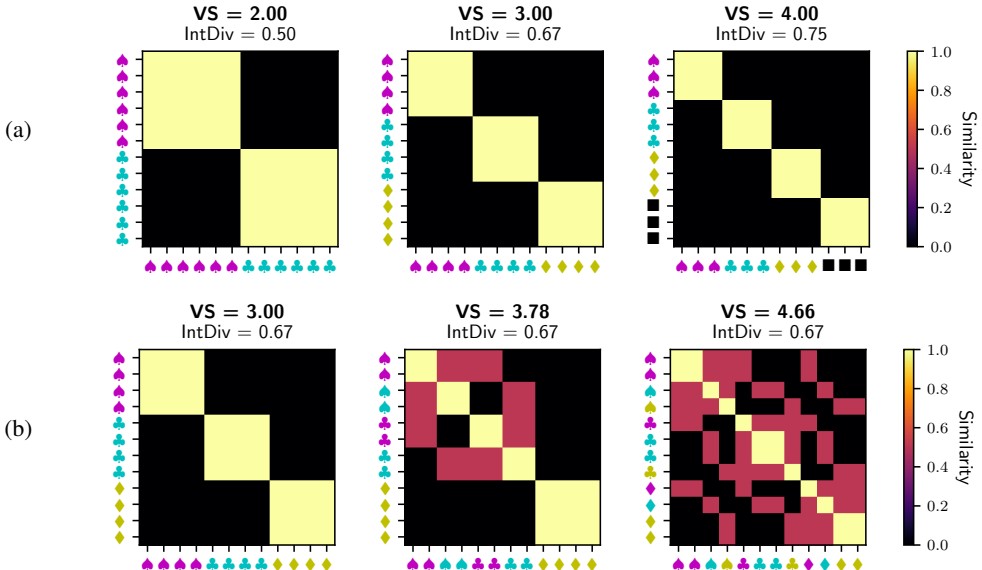

Figure 1: (a) The Vendi Score (VS) can be interpreted as the effective number of unique elements in a sample. It increases linearly with the number of modes in the dataset. IntDiv, the expected dissimilarity, becomes less sensitive as the number of modes increases, converging to 1. (b) IntDiv does not take into account correlations between features, but VS does. VS is highest when the items in the sample differ in many attributes, and the attributes are not correlated with each other.

**Contributions.** We summarize our contributions as follows:

- We extend ecological diversity to ML, and propose the Vendi Score, a metric for evaluating diversity in ML. We study the properties of the Vendi Score, which provides us with a more formal understanding of desiderata for diversity.

- We showcase the flexibility and wide applicability of the Vendi Score–characteristics that stem from its sole reliance on the sample to be evaluated for diversity and a user-defined similarity function–and highlight the shortcomings of existing metrics used to measure diversity in different domains.

## 2 RELATED WORK

**Are we measuring diversity correctly in ML?** Several existing metrics for diversity rely on a reference distribution/dataset, which hinders flexibility. These reference-based metrics define diversity in terms of coverage of a reference sample. They assume access to an embedding function that maps samples to real-valued vectors, such as a pretrained Inception model (Szegedy et al., 2016). The Fréchet Inception Distance (FID; Heusel et al., 2017) measures the Wasserstein-2 distance between Gaussian distributions fit to reference embeddings and sample embeddings. FID was originally proposed for evaluating image GANs but has been applied to text (Cífka et al., 2018) and molecules (Preuer et al., 2018) using domain-specific neural network encoders. Sajjadi et al. (2018) proposed a two-metric evaluation using precision and recall, with precision measuring quality and recall measuring diversity, in terms of coverage of the reference distribution, and a number of variations and improvements have been proposed (Kynkäänniemi et al., 2019; Simon et al., 2019; Naeem et al., 2020). Compared to these approaches, the Vendi Score can measure diversity without relying on a reference distribution/dataset.

Some other existing metrics evaluate diversity using a pre-trained classifier. Inception Score (IS; Salimans et al., 2016) evaluates diversity using the entropy of the marginal distribution of class labels predicted by an ImageNet classifier. More generally, given a pretrained classifier, mode coverage can be calculated directly by classifying test samples and calculating the number of unique modes or the

entropy of the mode distribution (Srivastava et al., 2017). These metrics define diversity in terms of predefined categories, and require knowledge of the ground truth categories and a separate classifier.

In some discrete domains, diversity is often evaluated in terms of the distribution of unique features. For example in NLP, a standard metric is n-gram diversity, which is defined as the number of distinct n-grams divided by the total number of n-grams (e.g. Li et al., 2016). These metrics require an explicit, discrete feature representation.

There are proposed metrics that use similarity scores to define diversity. The most widely used metric of this form is the average pairwise dissimilarity score. In biology, average dissimilarity is known as Internal Diversity (IntDiv; Benhenda, 2017), with similarity defined as the Jaccard (Tanimoto) similarity between molecular fingerprints. Given samples $x_1, \ldots, x_n$ and a pairwise similarity function $k$ taking values between 0 and 1, IntDiv is defined:

$$\text{IntDiv}(x_1, \ldots, x_n) = 1 - \frac{1}{n^2} \sum_{i,j} k(x_i, x_j).$$

In text, variants of this metric include Pairwise BLEU (Shen et al., 2019) and D-Lex-Sim (Fomicheva et al., 2020), in which the similarity function is an n-gram overlap metric such as BLEU (Papineni et al., 2002). As illustrated in Figure 1, IntDiv becomes less sensitive as diversity increases and does not account for correlations between features. Related to the metric we propose here is a similarity-sensitive diversity metric proposed in ecology by Leinster and Cobbold (2012), and which was introduced in the context of ML by Posada et al. (2020). This metric is based on a notion of entropy defined in terms of a *similarity profile*, a vector whose entries are equal to the expected similarity scores of each element. Like IntDiv, it does not account for correlations between features.

Some other diversity metrics in the ML literature fall outside of these categories. The Birthday Paradox Test (Arora and Zhang, 2018) aims to estimate the size of the support of a generative model, but requires some manual inspection of samples. GILBO (Alemi and Fischer, 2018) is a reference-free metric but is only applicable to latent variable generative models.

**Determinantal point processes.** The Vendi Score bears a relationship to determinantal point processes (DPPs), which have been used in machine learning for diverse subset selection (Kulesza et al., 2012). A DPP is a probability distribution over subsets of a ground set $\mathcal{X}$ parameterized by a positive semidefinite kernel matrix $\boldsymbol{K}$. The likelihood of drawing any subset $X \subseteq \mathcal{X}$ is defined as proportional to $|\boldsymbol{K}_X|$, the determinant of the similarity matrix restricted to elements in $X$: $p(X) \propto |\boldsymbol{K}_X| = \prod_i \lambda_i$, where $\lambda_i$ are the eigenvalues of $\boldsymbol{K}_X$. The likelihood function has a geometric interpretation, as the square of the volume spanned by the elements of $X$ in an implicit feature space. However, the DPP likelihood is not commonly used for evaluating diversity, and has some limitations. For example, it is always equal to 0 if the sample contains any duplicates, and the geometric meaning is arguably less straightforward to interpret than the Vendi Score, which can be understood in terms of the effective number of dissimilar elements.

**Spectral clustering.** The eigenvalues of the similarity matrix are also related to spectral clustering algorithms (Von Luxburg, 2007), which use a matrix known as the graph Laplacian, defined $\boldsymbol{L} = \boldsymbol{D} - \boldsymbol{K}$, where $\boldsymbol{K}$ is a symmetric, weighted adjacency matrix with non-negative entries, and $\boldsymbol{D}$ is a diagonal matrix with $D_{i,i} = \sum_j K_{i,j}$. The eigenvalues of $\boldsymbol{L}$ can be used to characterize different properties of the graph—for example, the multiplicity of the eigenvalue 0 is equal to the number of connected components. As a metric for diversity, the Vendi Score is somewhat more general than the number of connected components: it provides a meaningful measure even for fully connected graphs, and captures within-component diversity.

## 3 Measuring Diversity with the Vendi Score

We now define the Vendi Score, state its properties, and study its computational complexity. (We relegate all proofs of lemmas and theorems to the appendix.)

### 3.1 Defining the Vendi Score

To define a diversity metric in ML we look to ecology, the field that centers diversity in its work. In ecology, one main way diversity is defined is as the exponential of the entropy of the distribution of

the species under study (Jost, 2006; Leinster, 2021). This is a reasonable index for diversity. Consider a population with a uniform distribution over $n$ species, with entropy $\log(n)$. This population has maximal ecological diversity $n$, the same diversity as a population with $n$ members, each belonging to a different species. The ecological diversity decreases as the distribution over the species becomes less uniform, and is minimized and equal to one when all members of the population belong to the same species.

How can we extend this way of thinking about diversity to ML? One naive approach is to define diversity as the exponential of the Shannon entropy of the probability distribution defined by a machine learning model or dataset. However, this approach is limiting in that it requires a probability distribution for which entropy is tractable, which is not possible in many ML settings. We would like to define a diversity metric that only relies on the samples being evaluated for diversity. And we would like for such a metric to achieve its maximum value when all samples are dissimilar and its minimum value when all samples are the same. This implies the need to define a similarity function over the samples. Endowed with such a similarity function, we can define a form of entropy that only relies on the samples to be evaluated for diversity. This leads us to the Vendi Score:

**Definition 3.1** (Vendi Score). *Let $x_1, \ldots, x_n \in \mathcal{X}$ denote a collection of samples, let $k : \mathcal{X} \times \mathcal{X} \to \mathbb{R}$ be a positive semidefinite similarity function, with $k(x, x) = 1$ for all $x$, and let $\boldsymbol{K} \in \mathbb{R}^{n \times n}$ denote the kernel matrix with entry $K_{i,j} = k(x_i, x_j)$. Denote by $\lambda_1, \ldots, \lambda_n$ the eigenvalues of $\boldsymbol{K}/n$. The Vendi Score (VS) is defined as the exponential of the Shannon entropy of the eigenvalues of $\boldsymbol{K}/n$:*

$$VS_k(x_1, \ldots, x_n) = \exp\left(-\sum_{i=1}^{S} \lambda_i \log \lambda_i\right), \tag{1}$$

*where we use the convention $0 \log 0 = 0$.*

To understand the validity of the Vendi Score as a mathematical object, note that the eigenvalues of $\boldsymbol{K}/n$ are nonnegative (because $k$ is positive semidefinite) and sum to one (because the diagonal entries of $\boldsymbol{K}/n$ are equal to $1/n$). The Shannon entropy is therefore well-defined and the Vendi Score is well-defined. In this form, the Vendi Score can also be seen as the *effective rank* of the kernel matrix $\boldsymbol{K}/n$. Effective rank was introduced by Roy and Vetterli (2007) in the context of signal processing; the effective rank of a matrix is defined as the exponential of the entropy of the normalized singular values. The Vendi Score can be expressed directly as a function of the kernel similarity matrix $\boldsymbol{K}$:

**Lemma 3.1.** *Consider the same setting as Definition 3.1. Then*

$$VS_k(x_1, \ldots, x_n) = \exp\left(-\operatorname{tr}\left(\frac{\boldsymbol{K}}{n} \log \frac{\boldsymbol{K}}{n}\right)\right). \tag{2}$$

The lemma makes explicit the connection of the Vendi Score to quantum statistical mechanics: the Vendi Score is equal to the exponential of the von Neumann entropy associated with $\boldsymbol{K}/n$ (Bach, 2022).

Our formulation of the Vendi Score assumes that $x_1, \ldots, x_n$ were sampled independently, and so $p(x_i) \approx \frac{1}{n}$ for all $i$. This is the usual setting in ML and the setting we study in our experiments. However, we can generalize the Vendi Score to a setting in which we have an explicit probability distribution over the sample space $\mathcal{X}$ (see Definition 3.1 in the appendix).

### 3.2 Understanding the Vendi Score

The Vendi Score has several desirable properties as a diversity metric. We summarize them in the following theorem.

**Theorem 3.1** (Properties of the Vendi Score). *Consider the same definitions in Definition 3.1.*

(a) *Effective number: If $k(x_i, x_j) = 0$ for all $i \neq j$, then $VS_k(x_1, \ldots, x_n)$ is maximized and equal to $n$. If $k(x_i, x_j) = 1$ for all $i, j$, then $VS_k(x_1, \ldots, x_n)$ is minimized and equal to 1.*

(b) *Partitioning: Suppose $S_1, \ldots, S_m$ are collections of samples such that, for any $i \neq j$, for all $x \in S_i, x' \in S_j, k(x, x') = 0$. Then the diversity of the combined samples depends only on the diversities of $S_1, \ldots, S_m$ and their relative sizes. In particular, if $p_i = |S_i|/\sum_j |S_j|$ is the relative size of $S_i$ and $H(p_1, \ldots, p_m)$ denotes the Shannon entropy, then VS is given by the geometric mean, $VS_k(S_1, \ldots, S_m) = \exp(H(p_1, \ldots, p_m)) \prod_{i=1}^{m} VS_k(S_i)^{p_i}$.*

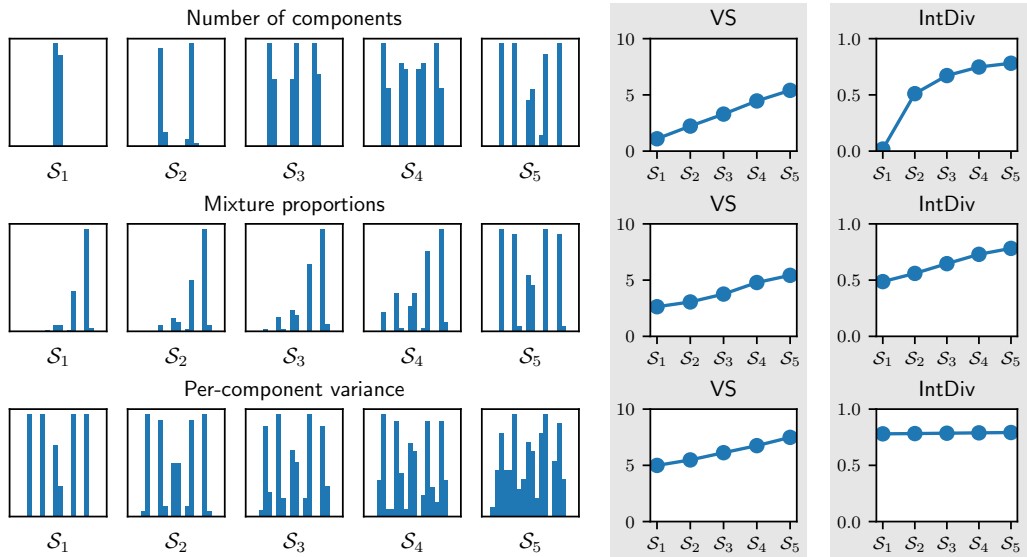

Figure 2: VS increases proportionally with diversity in three sets of synthetic datasets. In each row, we sample datasets from univariate mixture-of-normal distributions, varying either the number of components, the mixture proportions, or the per-component variance. The datasets are depicted in the left, as histograms, and the diversity scores are plotted on the right.

*(c) Symmetry: If $\pi_1, \ldots, \pi_n$ is a permutation of $1, \ldots, n$, then $VS_k(x_1, \ldots, x_n) = VS_k(x_{\pi_1}, \ldots, x_{\pi_n})$.*

Taking the exponential of entropy turns it into an *effective number*. The value of measuring diversity with effective numbers has been argued in ecology (e.g. Hill, 1973; Patil and Taillie, 1982; Jost, 2006) and economics (Adelman, 1969). Effective numbers provide a consistent basis for interpreting diversity scores, and make it possible to compare diversity scores using ratios and percentages. For example, suppose a population contains one million equally abundant species, and 50% go extinct. The diversity score will decrease by 50%, while the entropy will decrease by only about 5%.

### 3.3 CALCULATING THE VENDI SCORE

Calculating the Vendi Score for a sample of $n$ elements requires finding the eigenvalues of an $n \times n$ matrix, which has a time complexity of $O(n^3)$. The Vendi Score can be approximated using column sampling methods (i.e. the Nyström method; Williams and Seeger, 2000). However, in many of the applications we consider, the similarity function we use is the cosine similarity between explicit feature vectors $\phi(x) \in \mathbb{R}^d$, with $d < n$. That is, $\boldsymbol{K} = \boldsymbol{X}^\top \boldsymbol{X}$, where $\boldsymbol{X} \in \mathbb{R}^{n \times d}$ is the feature matrix with row $\boldsymbol{X}_{i,:} = \phi(x_i)/\|\phi(x_i)\|_2$. The eigenvalues of $\boldsymbol{K}/n$ are the same as the eigenvalues of the matrix $\boldsymbol{X}\boldsymbol{X}^\top/n$, therefore we can calculate the Vendi Score exactly in a time of $O(d^2 n + d^3) = O(d^2 n)$. This is the same complexity as existing metrics such as FID (Heusel et al., 2017), which require calculating the covariance matrix of Inception embeddings.

## 4 EXPERIMENTS

We illustrate the Vendi Score (VS) on synthetic data to illustrate that it captures intuitive notions of diversity, and then apply it to a variety of setting in ML. We used VS to evaluate the diversity of generative models of molecules, an application where diversity plays an important role in enabling discovery, and found that VS identifies some model weaknesses that are not detected by IntDiv, the standard metric in that domain. We also applied VS to generative models of images, and decoding algorithms of text, where we found it confirms what we know about diversity in those applications. We also used VS to measure mode collapse in GANs and show that it reveals finer-grained distinctions in diversity than current metrics for measuring mode collapse. Finally, we used VS to analyze the

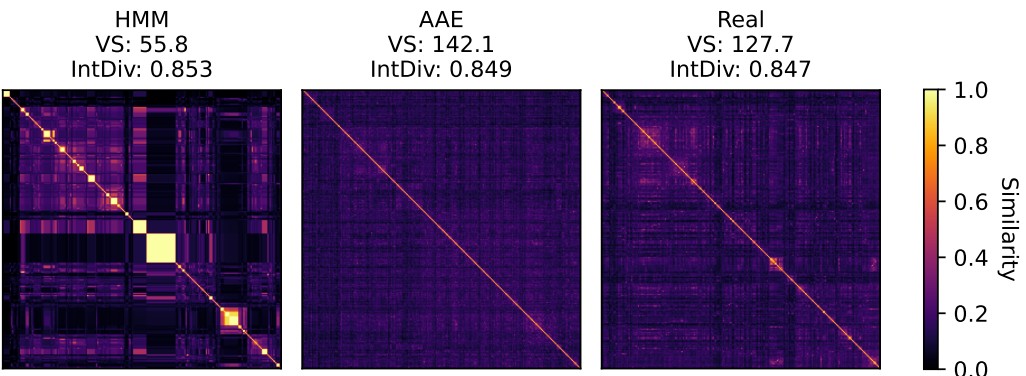

Figure 3: The kernel matrices for 250 molecules sampled from the HMM, AAE, and the original dataset, sorted lexicographically by SMILES string representation. The samples have similar IntDiv scores, but the HMM samples score much lower on VS. The figure shows that the HMM generates a number of exact duplicates, which are reflected by VS but not IntDiv.

diversity of several image, text, and molecule datasets, gaining insights into the least and most diverse elements of those datasets. (Implementation details are provided in Appendix B.)

### 4.1 SYNTHETIC EXPERIMENTS

To illustrate the behavior of the Vendi Score, we calculate the diversity of simple datasets drawn from a mixture of univariate normal distributions, varying either the number of components, the mixture proportions, or the per-component variance. We measure similarity using the RBF kernel: $k(x, x') = \exp(\|x - x'\|^2/2\sigma^2)$. The results are illustrated in Figure 2. VS behaves consistently and intuitively in all three settings: in each case, VS can be interpreted as the effective number of modes, ranging between one and five in the first two rows and increasing from five to seven in the third row as we increase within-mode variance. On the other hand, the behavior of IntDiv is different in each settings: for example, IntDiv is relatively insensitive to within-mode variance, and additional modes bring diminishing returns.

In Appendix C.1, we also validate that VS captures mode dropping in a simulated setting, using image and text classification datasets, where we have information about the ground truth class distribution. In both cases, VS has a stronger correlation with the true number of modes compared to IntDiv.

### 4.2 EVALUATING MOLECULAR GENERATIVE MODELS FOR DIVERSITY

Next, we evaluate the diversity of samples from generative models of molecules. For generative models to be useful for the discovery of novel molecules, they ought to be diverse. The standard diversity metric in this setting is IntDiv. We evaluate samples from generative models provided in the MOSES benchmark (Polykovskiy et al., 2020), using the first 2,500 valid molecules in each sample. Following prior work, our similarity function is the Morgan fingerprint similarity (radius 2), implemented in RDKit.[2] In Figure 3, we highlight an instance where IntDiv and VS disagree: IntDiv ranks the HMM among the most diverse models, while VS ranks it as the least diverse (the complete results are in Appendix Table 4). The HMM has a high IntDiv score because, on average, the HMM molecules have low pairwise similarity scores, but there are a number of clusters of identical or nearly identical molecules.

### 4.3 ASSESSING MODE COLLAPSE IN GENERATIVE ADVERSARIAL NETWORKS (GANS)

Mode collapse is a failure mode of GANs that has received a lot of attention from the ML community (Metz et al., 2017; Dieng et al., 2019). The main metric for measuring mode collapse, called *number of modes*, can only be used to assess mode collapse for GANs trained on a labelled dataset.

---

[2]RDKit: Open-source Cheminformatics. https://www.rdkit.org.

| Model | Modes | Mode Div. | VS |
|---|---|---|---|
| Self-cond. GAN | 1000 | 921.0 | 746.7 |
| PresGAN | 1000 | 948.7 | 866.6 |
| Original | 1000 | 950.8 | 943.7 |

Table 1: VS reveals that even GANs that capture all the modes of a labeled dataset can be less diverse than the original dataset, indicating that it captures a more fine-grained notion of diversity than number of modes.

| Model | IS↑ | FID↓ | Prec↑ | Rec↑ | VS↑ | Model | IS↑ | FID↓ | Prec↑ | Rec↑ | VS↑ |
|---|---|---|---|---|---|---|---|---|---|---|---|
| **CIFAR-10** | | | | | | **LSUN Bedroom 256×256** | | | | | |
| Original | | | | | 19.50 | Original | | | | | 8.99 |
| VDVAE | 5.82 | 40.05 | 0.63 | 0.35 | 12.87 | StyleGAN | 2.55 | 2.35 | 0.59 | 0.48 | 8.76 |
| DenseFlow | 6.01 | 34.54 | 0.62 | 0.38 | 13.55 | ADM | 2.38 | 1.90 | 0.66 | 0.51 | 7.97 |
| IDDPM | 9.24 | 4.39 | 0.66 | 0.60 | 16.86 | RQ-VT | 2.56 | 3.16 | 0.60 | 0.50 | 8.48 |
| **ImageNet 64×64** | | | | | | **LSUN Cat 256×256** | | | | | |
| Original | | | | | 43.93 | Original | | | | | 15.12 |
| VDVAE | 9.68 | 57.57 | 0.47 | 0.37 | 18.04 | StyleGAN2 | 4.84 | 7.25 | 0.58 | 0.43 | 13.55 |
| DenseFlow | 5.62 | 102.90 | 0.36 | 0.17 | 12.71 | ADM | 5.19 | 5.57 | 0.63 | 0.52 | 13.09 |
| IDDPM | 15.59 | 19.24 | 0.59 | 0.58 | 24.28 | RQ-VT | 5.76 | 10.69 | 0.53 | 0.48 | 14.91 |

Table 2: VS generally agrees with the existing metrics. On low-resolution datasets (left) the diffusion model performs better on all of the metrics. On the LSUN datasets (right), the diffusion model gets the best quality scores as measured by FID, but scores lower on VS. No model matches the diversity score of the original dataset they were trained on.

Number of modes is computed by training a classifier on the labeled training data and counting the number of unique classes that are predicted by the trained classifier for the generated samples. In Table 1, we evaluate two models that were trained on the StackedMNIST dataset, a standard setting for evaluating mode collapse in GANs. StackedMNIST is created by stacking three MNIST images along the color channel, creating 1000 classes corresponding to 1000 number of modes.

We calculate VS using the probability product kernel (Jebara et al., 2004): $k(x, x') = \sum_y p(y \mid x)^{\frac{1}{2}} p(y \mid x')^{\frac{1}{2}}$, where the class likelihoods are given by the classifier. We compare PresGAN (Dieng et al., 2019) and Self-conditioned GAN (Liu et al., 2020), two GANs that are known to capture all the modes. Table 1 shows that PresGAN and Self-conditioned GAN have the same diversity according to number of modes, they capture all 1000 modes. However, VS reveals a more fine-grained notion of diversity, indicating that PresGAN is more diverse than Self-conditioned GAN and that both are less diverse than the original dataset. One possibility is that VS is capturing imbalances in the mode distribution. To see whether this is the case, we also calculate Mode Diversity, the exponential entropy of the predicted mode distribution: $\exp H(\hat{p}(y))$, where $\hat{p}(y) = \frac{1}{n} \sum_{i=1}^{n} p(y \mid x_i)$. The generative models score lower on VS than Mode Diversity, indicating that low scores cannot be entirely attributed to imbalances in the mode distribution. Therefore VS captures more aspects of diversity, even when we are using the same representations as existing methods.

### 4.4 Evaluating image generative models for diversity

We now evaluate several recent models for unconditional image generation, comparing the diversity scores with standard evaluation metrics, Inception Score (IS; Salimans et al., 2016), Frechet Inception Distance (FID; Heusel et al., 2017), Precision (Sajjadi et al., 2018), and Recall (Sajjadi et al., 2018). The models we evaluate represent popular classes of generative models, including a variational autoencoder (VDVAE; Child, 2020), a flow model (DenseFlow; Grcić et al., 2021), diffusion models (IDDPM, Nichol and Dhariwal, 2021; ADM Dhariwal and Nichol, 2021), GAN-based models (Karras et al., 2019; 2020), and an auto-regressive model (RQ-VT; Lee et al., 2022). The models are trained on CIFAR-10 (Krizhevsky, 2009), ImageNet (Russakovsky et al., 2015), or two

categories from the LSUN dataset (Yu et al., 2015). We either select models that provide precomputed samples, or download publicly available model checkpoints and sample new images using the default hyperparameters. (More details are in Appendix B.)

The standard metrics in this setting use a pre-trained Inception ImageNet classifier to map images to real vectors. Therefore, we calculate VS using the cosine similarity between Inception embeddings, using the same 2048-dimensional representations used for evaluating FID and Precision/Recall. As a result, the highest possible similarity score is 2048. The baseline metrics are reference-based, with the exception of the IS. FID and IS capture diversity implicitly. Recall was introduced to capture diversity explicitly, with diversity defined as coverage of the reference distribution.

The results of this comparison are in Table 2. On the lower resolution datasets (left), VS generally agrees with the existing metrics. On those datasets the diffusion model performs better on all of the metrics. On the LSUN datasets (right), the diffusion model gets the best quality scores as measured by FID, but scores lower on VS. No model matches the diversity score of the original dataset they were trained on. In addition to comparing the diversity of the models, we can also compare the diversity scores between datasets: as a function of Inception similarity, the most diverse dataset is ImageNet 64×64, followed by CIFAR-10, followed by LSUN Cat, and then LSUN Bedroom. This agrees with the intuition that the Inception network will induce the highest-rank embedding space for datasets that most resemble the training distribution.

VS should be understood as the diversity with respect to a specific similarity function, in this case, the Inception ImageNet similarity. We illustrate this point in the appendix (Figure 6) by comparing the top eigenvalues of the kernel matrices corresponding to the cosine similarity between Inception embeddings and pixel vectors. Inception similarity captures a form of semantic similarity, with components corresponding to particular cat breeds, while the pixel kernel provides a simple form of visual similarity, with components corresponding to broad differences in lightness, darkness, and color.

## 4.5 EVALUATING DECODING ALGORITHMS FOR TEXT FOR DIVERSITY

| Source | BLEU-4 | N-gram div. | VS |
|---|---|---|---|
| Human | | 0.82 | 4.88 |
| Beam Search | 0.27 | 0.42 | 3.00 |
| DBS $\gamma = 0.2$ | 0.25 | 0.49 | 3.16 |
| DBS $\gamma = 0.5$ | 0.22 | 0.63 | 4.14 |
| DBS $\gamma = 0.8$ | 0.21 | 0.68 | 4.37 |

Table 3: Quality and diversity scores for an image captioning model using Beam Search or Diverse Beam Search (DBS). Increasing the diversity penalty $\gamma$ leads to higher diversity scores but a lower quality score (measured by BLEU-4).

We evaluate diversity on the MS COCO image-captioning dataset (Lin et al., 2014), following prior work on diverse text generation (Vijayakumar et al., 2018). In this setting, the subjects of evaluation are diverse decoding algorithms rather than parametric models. Given a fixed model $p(x \mid c)$, where $c$ is some conditioning context, the aim is to identify a "Diverse N-Best List", a list of sentences that have high likelihood but are mutually distinct. a list of sentences that have high likelihood but are mutually distinct. The baseline metric we compare to is n-gram diversity (Li et al., 2016), which is the proportion of unique n-grams divided by the total number of n-grams. We define similarity using the n-gram overlap kernel: for a given $n$, the n-gram kernel $k_n$ is the cosine similarity between bag-of-n-gram feature vectors. We use the average of $k_1, \ldots, k_4$. This ensures that VS and n-gram diversity are calculated using the same feature representation. Each image in the validation split has five captions written by different human annotators, and we compare these with captions generated by a publicly available captioning model trained on this dataset. [3] For each image, we generate five captions using either beam search or diverse beam search (DBS Vijayakumar et al., 2018). DBS takes a parameter, $\gamma$, called the diversity penalty, and we vary this between 0.2, 0.6, and 0.8.

---

[3]https://huggingface.co/ydshieh/vit-gpt2-coco-en-ckpts

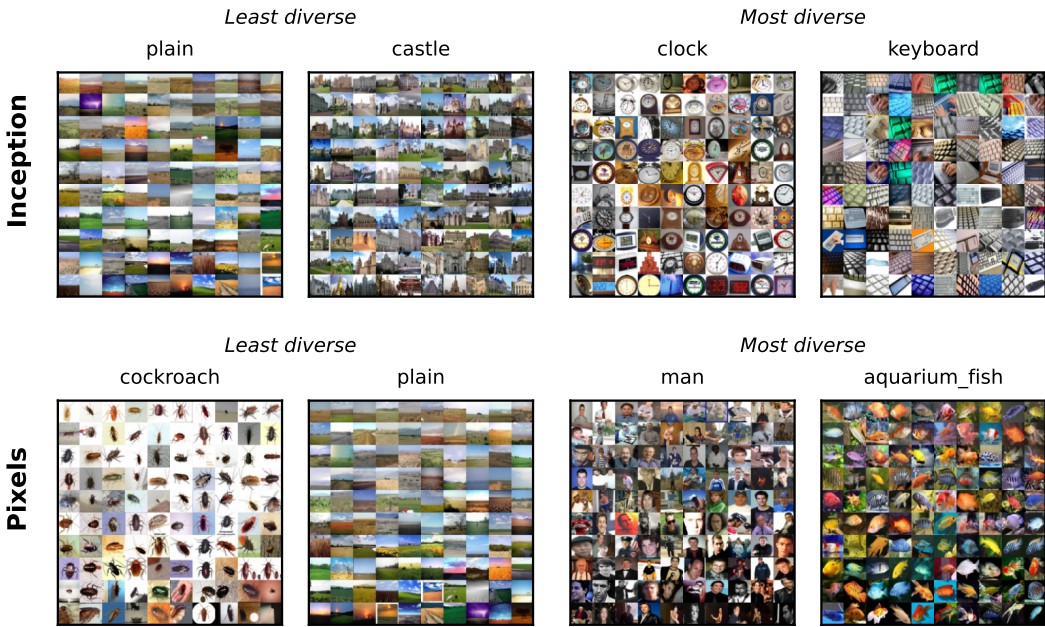

Figure 4: The categories in CIFAR-100 with the lowest and highest VS, defining similarity as the cosine similarity between either Inception embeddings or pixel vectors. We show 100 examples from each category, in decreasing order of average similarity, with the image at the top left having the highest average similarity scores according to the corresponding kernel.

Table 3 shows that all diversity metrics increase as expected, ranking beam search the lowest, the human captions the highest, and DBS in between, increasing with the diversity penalty. The human diversity score of 4.88 can be interpreted as meaning that, on average, all five human-written captions are almost completely dissimilar from each other, while beam search effectively returns only three distinct responses for every five that it generates.

### 4.6 DIAGNOSING DATASETS FOR DIVERSITY

In Figure 4, we calculate VS for samples from different categories in CIFAR-100, using the cosine similarity between either Inception embeddings or pixel vectors. The pixel diversity is highest for categories like "aquarium fish", which vary in color, brightness, and orientation, and lowest for categories like "cockroach" in which images have similar regions of high pixel intensity (like white backgrounds). The Inception diversity is less straightforward to interpret, but might correspond to some form of semantic diversity—for example, the Inception diversity might be lower for classes like "castle," that correspond to distinct ImageNet categories, and higher for categories like "clock" and "keyboard" that are more difficult to classify. In Appendix C.5, we show additional examples from text, molecules, and other image datasets.

## 5 DISCUSSION

We introduced the Vendi Score, a metric for evaluating diversity in machine learning. The Vendi Score is defined as a function of the pairwise similarity scores between elements of a sample and can be interpreted as the effective number of unique elements in the sample. The Vendi Score is interpretable, general, and applicable to any domain where similarity can be defined. It is unsupervised, in that it doesn't require labels or a reference probability distribution. Importantly, the Vendi Score allows its user to specify the form of diversity they want to measure via the similarity function. We showed the Vendi Score can be computed efficiently exactly and showcased its usefulness in several ML applications, different datasets, and different domains. In future work, we'll leverage the Vendi Score to improve data augmentation, an important ML approach in settings with limited data.

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

## A   PROOFS

### A.1   PROBABILITY-WEIGHTED VENDI SCORE

**Definition A.1** (Probability-Weighted Vendi Score). *Let $\boldsymbol{p} \in \Delta_n$ denote a probability distribution on a discrete space $\mathcal{X} = \{x_1, \dots, x_n\}$, where $\Delta_n$ denotes the $(n-1)$-dimensional simplex, let $k : \mathcal{X} \times \mathcal{X} \to \mathbb{R}$ be a positive semidefinite similarity function, with $k(x, x) = 1$ for all $x$, and let $\boldsymbol{K} \in \mathbb{R}^{n \times n}$ denote the kernel matrix with $K_{i,j} = k(x_i, x_j)$. Let $\tilde{\boldsymbol{K}}_{\boldsymbol{p}} = \operatorname{diag}(\sqrt{\boldsymbol{p}}) \boldsymbol{K} \operatorname{diag}(\sqrt{\boldsymbol{p}})$ denote the probability-weighted kernel matrix. Let $\lambda_1, \dots, \lambda_n$ denote the eigenvalues of $\tilde{\boldsymbol{K}}_{\boldsymbol{p}}$. The Vendi Score (VS) is defined as the exponential of the Shannon entropy of the eigenvalues of $\tilde{\boldsymbol{K}}_{\boldsymbol{p}}$:*

$$VS_k(x_1, \dots, x_n, \boldsymbol{p}) = \exp\left(-\sum_{i=1}^{S} \lambda_i \log \lambda_i\right). \tag{3}$$

When all elements in the sample are completely dissimilar, the probability-weighted Vendi Score defined in Definition A.1 reduces to the exponential of the Shannon entropy of the weighting distribution:

**Lemma A.1.** *Let $\boldsymbol{p} \in \Delta_n$ be a probability distribution over $x_1, \dots, x_n$ and suppose $k(x_i, x_j) = 0$ for all $i \neq j$. Then $VS_k(x_1, \dots, x_n, \boldsymbol{p}) = \exp H(\boldsymbol{p})$, the exponential of the Shannon entropy of $\boldsymbol{p}$.*

### A.2   PROOF OF ??

Rather than prove **??** directly, we prove a more general lemma for the probability-weighted kernel matrix $\tilde{\boldsymbol{K}}_{\boldsymbol{p}}$ defined in Definition A.1. **??** follows by setting $\boldsymbol{p} = (1/N, \dots, 1/N)$ to be the uniform distribution on $N$ elements.

**Lemma A.2.** *Let $\boldsymbol{K} \in \mathbb{R}^{N \times N}$ denote a positive semi-definite kernel matrix with $\boldsymbol{K}_{ii} = 1$ for $i \in \{1, \dots, N\}$. Let $\Delta_N$ denote the $(N-1)$-dimensional simplex; its elements are vectors of dimension $N$ whose entries are nonnegative and sum to one. Let $\boldsymbol{p} = (p_1, \dots, p_n) \in \Delta_N$ denote a probability distribution on $N$ elements, and let $\boldsymbol{\lambda} = (\lambda_1, \dots, \lambda_N)$ denote the vector of eigenvalues of the probability-weighted kernel matrix $\tilde{\boldsymbol{K}}_{\boldsymbol{p}} = \operatorname{diag}(\sqrt{\boldsymbol{p}}) \boldsymbol{K} \operatorname{diag}(\sqrt{\boldsymbol{p}})$. Then $\boldsymbol{\lambda} \in \Delta_N$.*

*Proof.* $\boldsymbol{\lambda}$ is in the $(N-1)$-simplex if its entries are nonnegative and sum to one. The entries of $\boldsymbol{\lambda}$ are nonnegative because the eigenvalues of a positive semi-definite matrix are nonnegative, and $\tilde{\boldsymbol{K}}_{\boldsymbol{p}}$ is the product of positive semi-definite matrices and so is positive semi-definite iteslf. $(\operatorname{diag}(\sqrt{\boldsymbol{p}})$ is positive semi-definite because it is a diagonal matrix with nonnegative entries.) The entries sum to one because the trace of a square matrix is equal to the sum of its eigenvalues, and $\operatorname{tr}(\tilde{\boldsymbol{K}}_{\boldsymbol{p}}) = \sum_{i=1}^{N} \boldsymbol{K}_{ii} p_i = 1$. $\square$

### A.3   PROOF OF LEMMA 3.1

*Lemma.* Consider the same setting as Definition 3.1. Then

$$VS_k(x_1, \dots, x_n) = \exp\left(-\operatorname{tr}\left(\frac{\boldsymbol{K}}{n} \log \frac{\boldsymbol{K}}{n}\right)\right). \tag{4}$$

*Proof.* For any square matrix $\boldsymbol{X} \in \mathbb{R}^{n \times n}$, if $\boldsymbol{X}$ has an eigendecomposition $\boldsymbol{X} = \boldsymbol{U} \boldsymbol{\Lambda} \boldsymbol{U}^{-1}$, then $\log \boldsymbol{X} = \boldsymbol{U} (\log \boldsymbol{\Lambda}) \boldsymbol{U}^{-1}$, where $\log \boldsymbol{\Lambda} = \operatorname{diag}(\log \lambda_1, \dots, \log \lambda_n)$ is a diagonal matrix whose diagonal entries are the logarithms of the eigenvalues of $\boldsymbol{X}$. Also, $\operatorname{tr}(\boldsymbol{X}) = \operatorname{tr}\left(\boldsymbol{U} \boldsymbol{\Lambda} \boldsymbol{U}^{-1}\right) = \operatorname{tr}(\boldsymbol{\Lambda})$,

because the trace is similarity-invariant. $\boldsymbol{K}/n$ is diagonalizable because it is positive semidefinite, so let $\boldsymbol{K}/n = \boldsymbol{U}\boldsymbol{\Lambda}\boldsymbol{U}^{-1}$ denote the eigendecomposition. Then

$$
\begin{aligned}
\operatorname{tr}(\boldsymbol{K}/n \log \boldsymbol{K}/n) &= \operatorname{tr}\left(\boldsymbol{U}\boldsymbol{\Lambda}\boldsymbol{U}^{-1} \log\left(\boldsymbol{U}\boldsymbol{\Lambda}\boldsymbol{U}^{-1}\right)\right) \\
&= \operatorname{tr}\left(\boldsymbol{U}\boldsymbol{\Lambda}\boldsymbol{U}^{-1}\boldsymbol{U}\left(\log \boldsymbol{\Lambda}\right)\boldsymbol{U}^{-1}\right) \\
&= \operatorname{tr}\left(\boldsymbol{\Lambda} \log \boldsymbol{\Lambda}\right) \\
&= \sum_{i=1}^{n} \lambda_i \log \lambda_i.
\end{aligned}
$$

Therefore

$$
\mathrm{VS}_k(x_1,\ldots,x_n) = \exp\left(-\sum_{i=1}^{n} \lambda_i \log \lambda_i\right) = \exp\left(-\operatorname{tr}\left(\frac{\boldsymbol{K}}{n}\log\frac{\boldsymbol{K}}{n}\right)\right).
$$

$\square$

### A.4 Proof of Lemma A.1

*Lemma.* Let $\boldsymbol{p} \in \Delta_n$ be a probability distribution over $x_1,\ldots,x_n$ and suppose $k(x_i, x_j) = 0$ for all $i \neq j$. Then $\mathrm{VS}_k(x_1,\ldots,x_n,\boldsymbol{p}) = \exp H(\boldsymbol{p})$, the exponential of the Shannon entropy of $\boldsymbol{p}$.

*Proof.* If all element in $\boldsymbol{p}$ are completely dissimilar, then $\tilde{\boldsymbol{K}}_{\boldsymbol{p}}$ is a diagonal matrix, and the eigenvalues $\lambda_1,\ldots,\lambda_S$ are the diagonal entries, which are the entries of $\boldsymbol{p}$. So the von Neumann entropy of $\tilde{\boldsymbol{K}}_{\boldsymbol{p}}$ is identical to the Shannon entropy of $\boldsymbol{p}$, and the exponential is the Vendi Score. $\square$

### A.5 Proof of Theorem 3.1

*Proof.* (a) Effective number: If $\boldsymbol{p}$ is the uniform distribution over $N$ completely dissimilar elements, then $\tilde{\boldsymbol{K}}_{\boldsymbol{p}}$ is a diagonal matrix with each diagonal entry equal to $1/N$. The eigenvalues of a diagonal matrix are the diagonal entries, so $\mathrm{VS}_K(\boldsymbol{p}) = \exp H(1/N,\ldots,1/N) = \exp \log N = N$. On the other hand, if all elements are completely similar to each other, then $\tilde{\boldsymbol{K}}_{\boldsymbol{p}}$ has rank one and so the Vendi Score is equal to one.

(b) Identical elements: The eigenvalues of $\tilde{\boldsymbol{K}}_{\boldsymbol{p}}$ are the same as the eigenvalues of the covariance matrix of the corresponding feature space:

$$
\tilde{\boldsymbol{\Sigma}}_{\boldsymbol{p}} = \sum_{i=1}^{N} p(x_i)\phi(x_i)\phi(x_i)^{\top}.
$$

Suppose elements $i$ and $j$ are identical, and let $\boldsymbol{p}'$ denote the probability distribution created by combining $i$ and $j$, i.e. $p_i' = p_i + p_j$ and $p_j' = 0$. Clearly, $\tilde{\boldsymbol{\Sigma}}_{\boldsymbol{p}} = \tilde{\boldsymbol{\Sigma}}_{\boldsymbol{p}'}$, and so $\mathrm{VS}_k(x_1,\ldots,x_n,\boldsymbol{p}) = \mathrm{VS}_k(x_1,\ldots,x_n,\boldsymbol{p}')$.

(c) Partitioning: Suppose $N$ samples are partitioned into $M$ groups $\mathcal{S}_1,\ldots,\mathcal{S}_M$ such that, for any $i \neq j$, for all $x \in S_i, x' \in S_j$, $k(x, x') = 0$. Let $p_i = |S_i|/\sum_j |S_j|$ denote the relative size of group $i$, and let $\boldsymbol{K}$ denote kernel matrix of $\cup_i S_i$, sorted in order of group index, and let $\boldsymbol{K}_{S_i}$ denote the restriction of $\boldsymbol{K}$ to elements in $S_i$. Then $\boldsymbol{K}/N$ is a block diagonal matrix, with each block $i$ equal to $p_i \boldsymbol{K}_{S_i}$. The eigenvalues of a block diagonal matrix are the combined eigenvalues of each block, and the partitioning property then follows from the partitioning property of the Shannon entropy.

(e) Symmetry: The eigenvalues of a matrix are unchanged by orthonormal transformation, and the Shannon entropy is symmetric in its arguments, so the Vendi Score is symmetric. $\square$

## B    Implementation Details

### B.1    Images

**Stacked MNIST**    We train GANs on Stacked MNIST using the publicly available code for Pres-GANs [4] and self-conditioned GANs [5]. The models share the same DCGAN (Radford et al., 2015) architecture and are trained on the same dataset of 60,000 Stacked MNIST images, rescaled to $32\times32$ pixels, and other hyperparameters are set according to the descriptions in the papers. The models are trained for 50 epochs and the diversity scores are evaluated every five epochs by taking 10,000 samples. For both models, we report the scores from the epoch corresponding to the highest VS score. As in prior work (Metz et al., 2017), we classify Stacked MNIST digits by applying a pretrained MNIST classifier to each color channel independently. The 1000-dimensional Stacked MNIST probability vector is then the tensor product of the three 10-dimensional probability vectors predicted for the three channels.

**Obtaining Image Samples**    In Section 4.4, we calculate the diversity scores of several recent generative models of images. We select models that represent a range of families of generative models and and provide publicly available samples or model checkpoints for common image datasets. On the low-resolution datasets, we generate 50,000 samples from each model using the official code for VDVAE,[6] DenseFlow,[7] and IDDPM,[8] each of which provides a checkpoint for unconditional image generation models on CIFAR-10 and ImageNet-64. For IDDPM, we sample using DDIM (Song et al., 2021) for 250 steps, and otherwise use the default sampling parameters. For the higher-resolution datasets, we use the 50,000 precomputed samples provided by Dhariwal and Nichol (2021)[9] for ADM and StyleGAN models. We obtain 50,000 samples from the RQ-VAE/Transformer model using the code and checkpoints provided by the authors,[10] with the default sampling parameters.

**Calculating Image Metrics**    In Table 2, we calculate standard image quality and diversity metrics, which are based on Inception embeddings. These Inception-based metrics are sensitive to a number of implementation details (Parmar et al., 2022) and in general cannot be compared directly between papers. For a consistent comparison, we calculate all scores using the evaluation code provided by Dhariwal and Nichol (2021). We also calculated FID and Precision/Recall using the provided reference images and statistics, with the exception of CIFAR-10, for which we use the training set as the reference. (The diversity scores of the *Original* datasets in Table 2 are calculated using these reference images.) As a result, the numbers in this table may not be directly comparable to results reported in prior work.

### B.2    Text

**Obtaining Image Captions**    In Section 4.5, we sample image captions from a pretrained image-captioning model,[11] which is publicly available in Hugging Face (Wolf et al., 2019), and we use the Hugging Face implementation of beam search and diverse beam search. For beam search we use a beam size of 5. For diverse beam search, we use a beam size of 10, a beam group size of 10, and set the number of return sequences to 5.

**Calculating Text Metrics**    The text metrics we use are calculated in terms of word n-grams, and therefore depend on how sentences are tokenized into words. We calculate all text metrics using the pre-trained wordpiece tokenizer used by the captioning models. We use the implementation of the BLEU score in NLTK (Bird, 2006).

---

[4]https://github.com/adjidieng/PresGANs

[5]https://github.com/stevliu/self-conditioned-gan

[6]https://github.com/openai/vdvae/

[7]https://github.com/matejgrcic/DenseFlow

[8]https://github.com/openai/improved-diffusion

[9]https://github.com/openai/guided-diffusion

[10]https://github.com/kakaobrain/rq-vae-transformer

[11]https://huggingface.co/ydshieh/vit-gpt2-coco-en-ckpts

## C    ADDITIONAL RESULTS

### C.1    ASSESSING MODE DROPPING IN DATASETS

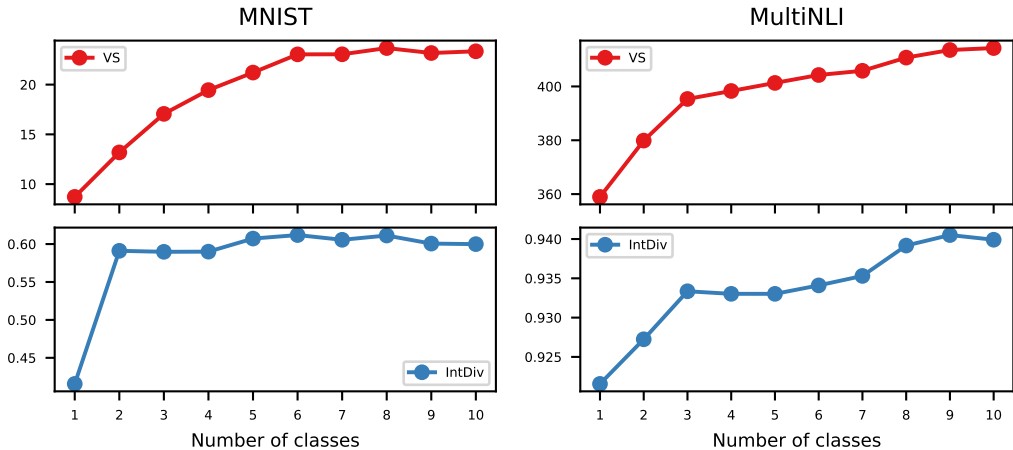

Figure 5: We evaluate VS and IntDiv on datasets containing 500 examples drawn uniformly from between one and ten classes: digits in MNIST and sentences genres in MultiNLI. Compared to IntDiv, VS increases more consistently with the number of classes.

In Figure 5, we examine whether VS captures mode dropping in a controlled setting, where we have information about the ground truth class distribution. We simulate mode dropping by sampling equal-sized subsets of two classification datasets, with each subset $\mathcal{S}_i$ containing examples sampled uniformly from the first $i$ categories. We perform this experiment one image dataset (MNIST) and one text dataset (MultiNLI; Williams et al., 2018), using simple similarity functions. For MNIST, we use the cosine similarity between pixel vectors. In MultiNLI, we use the premise sentences from the validation split (mismatched), which are drawn from one of ten genres. We define similarity using the n-gram overlap kernel: for a given $n$, the n-gram kernel $k_n$ is the cosine similarity between bag-of-n-gram feature vectors, and we use the average of $k_1, \ldots, k_4$.

The results (Figure 5) show that VS generally increases with the number of classes. In MNIST (left), VS increases roughly linearly for the first six digits (0-5) and then fluctuates. This could occur if the new modes are similar to the other modes in the sample, or have low internal diversity. In MultiNLI (right), VS increases monotonically with the number of genres represented in the sample. In both cases, VS has a stronger correlation with the number of modes compared to IntDiv.

### C.2    EVALUATING MOLECULAR GENERATIVE MODELS FOR DIVERSITY

We evaluate samples from generative models provided in the MOSES benchmark (Polykovskiy et al., 2020), using the first 2,500 valid molecules in each sample. Following prior work, our similarity function is the Morgan fingerprint similarity (radius 2), implemented in RDKit.[12] IntDiv ranks the HMM among the most diverse models, while VS ranks it as the least diverse (see Section 4.2).

### C.3    EVALUATING IMAGE GENERATIVE MODELS FOR DIVERSITY

In Table 5, we replicate the table described in Section 4.4 and add an additional column, which evaluates diversity using the cosine similarity between pixel vectors as the similarity function.

VS should be understood as the diversity with respect to a specific similarity function, in this case, the Inception ImageNet similarity. We illustrate this point in Figure 6 by comparing the top eigenvalues of the kernel matrices corresponding to the Inception similarity and the pixel similarity, which we calculate by resizing the images to $32{\times}32$ pixels and taking the cosine similarity between pixel

---

[12]RDKit: Open-source Cheminformatics. https://www.rdkit.org.

| Model | IntDiv | VS |
|---|---|---|
| Original | 0.855 | 403.9 |
| AAE | 0.859 | 501.1 |
| Char-RNN | 0.856 | 482.4 |
| Combinatorial | 0.873 | 536.9 |
| HMM | 0.871 | 250.9 |
| JTN | 0.856 | 489.5 |
| Latent GAN | 0.857 | 486.4 |
| N-gram | 0.874 | 479.8 |
| VAE | 0.856 | 475.3 |

Table 4: IntDiv and VS for generative models of molecules. The HMM has one of the highest IntDiv scores, but scores much lower on VS. An analysis of 250 molecules from the HMM reveals VS is more accurate in this case. (See Figure 3.)

| Model | IS↑ | FID↓ | Prec↑ | Rec↑ | $VS_I$↑ | $VS_P$↑ |
|---|---|---|---|---|---|---|
| **CIFAR-10** | | | | | | |
| Original | | | | | 19.50 | 3.52 |
| VDVAE | 5.82 | 40.05 | 0.63 | 0.35 | 12.87 | 3.34 |
| DenseFlow | 6.01 | 34.54 | 0.62 | 0.38 | 13.55 | 2.94 |
| IDDPM | 9.24 | 4.39 | 0.66 | 0.60 | 16.86 | 3.27 |
| **ImageNet 64×64** | | | | | | |
| Original | | | | | 43.93 | 4.43 |
| VDVAE | 9.68 | 57.57 | 0.47 | 0.37 | 18.04 | 4.24 |
| DenseFlow | 5.62 | 102.90 | 0.36 | 0.17 | 12.71 | 3.51 |
| IDDPM | 15.59 | 19.24 | 0.59 | 0.58 | 24.28 | 4.57 |

| Model | IS↑ | FID↓ | Prec↑ | Rec↑ | $VS_I$↑ | $VS_P$↑ |
|---|---|---|---|---|---|---|
| **LSUN Bedroom 256×256** | | | | | | |
| Original | | | | | 8.99 | 3.10 |
| StyleGAN | 2.55 | 2.35 | 0.59 | 0.48 | 8.76 | 3.09 |
| ADM | 2.38 | 1.90 | 0.66 | 0.51 | 7.97 | 3.27 |
| RQ-VT | 2.56 | 3.16 | 0.60 | 0.50 | 8.48 | 3.67 |
| **LSUN Cat 256×256** | | | | | | |
| Original | | | | | 15.12 | 4.58 |
| StyleGAN2 | 4.84 | 7.25 | 0.58 | 0.43 | 13.55 | 4.53 |
| ADM | 5.19 | 5.57 | 0.63 | 0.52 | 13.09 | 4.81 |
| RQ-VT | 5.76 | 10.69 | 0.53 | 0.48 | 14.91 | 5.83 |

Table 5: We evaluate samples from several recent models, measuring similarity using either Inception representations ($VS_I$) or pixels ($VS_P$). The pixel similarity score is the cosine similarity between pixel vectors, calculated after resizing the images to 32×32 pixels. The pixel similarity and Inception similarity scores do not always agree—for example, if the images in a sample represent a variety of ImageNet classes by share a similar color palette, we might expect the sample to have high Inception diversity but low pixel diversity. The pixel diversity scores are on a lower scale, indicating that this similarity metric is less capable of making fine-grained distinctions between the images in these samples.

vectors. Inception similarity provides a form of semantic similarity, with components corresponding to particular cat breeds, while the pixel kernel provides a simple form of visual similarity, with components corresponding to broad differences in lightness, darkness, and color.

## C.4 EVALUATING DECODING ALGORITHMS FOR TEXT FOR DIVERSITY

In Figure 7, we plot the relationship between VS and n-gram diversity using the MS-COCO captioning data and the n-gram overlap kernel described in Section 4.5. The figure shows that VS is highly correlated with n-gram diversity, which is expected given that our similarity function is based on n-gram overlap. Nonetheless, there are some data points that the metrics rank differently. This is because n-gram diversity conflates two properties: the diversity of n-grams within a single sentences and the n-gram overlap between sentences. We highlight two examples in Figure 8. In general, the instances that n-gram diversity ranks lower compared to VS contain individual sentences that repeat phrases. On the other hand, n-gram diversity can be inflated in cases when one sentence in the sample is much longer than the others, even if the other sentences are not diverse.

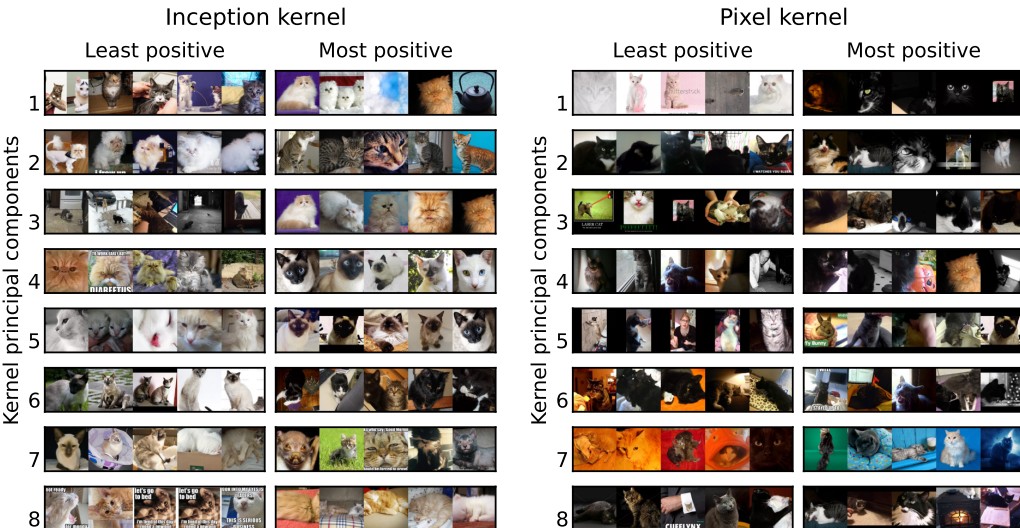

Figure 6: The choice of similarity function provides a way of specifying the notion of diversity that is relevant for a given application. We project LSUN Cat images along the top eigenvectors of the kernel matrix, using either Inception features or pixels to define similarity. Inception similarity provides a form of semantic similarity, with components corresponding to particular cat breeds, while the pixel kernel captures visual similarity. For each eigenvector $u$, we show the four images with the highest and lowest entries in $u$. For both kernels, every similarity score is positive, so all entries in the top eigenvector have the same sign; the images with the heighest weights in this component have the highest expected similarity scores. The remaining eigenvectors partition the images along different dimensions of variation.

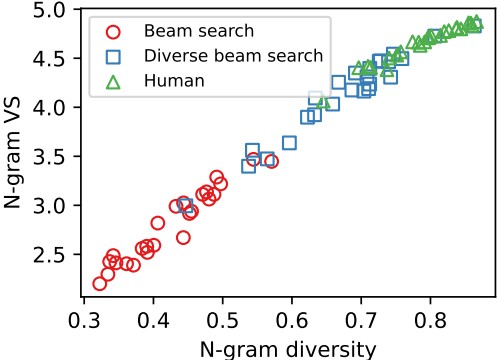

Figure 7: VS is correlated with N-gram diversity. Each point represents a group of five captions for a particular image.

### C.5 DIAGNOSING DATASETS FOR DIVERSITY

**Molecules**    We evaluate the diversity scores of molecules in the GoodScents database of perfume materials,[13] which has been used in prior machine learning research on odor modeling (Sanchez-Lengeling et al., 2019). We use the standardized version of the data provided by the Pyrfume library. [14]   Each molecule in the dataset is labeled with one or more odor descriptors (for example, "clean, oily, waxy" or "floral, fruity, green"). We form groups of molecules corresponding to the seven most common odor descriptors, with each group consisting of 500 randomly sampled

---

[13]http://www.thegoodscentscompany.com/
[14]https://pyrfume.org/

High Vendi Score, low n-gram diversity:

- *two men in bow ties standing next to steel rafter.*
- *several men in suits talking together in a room.*
- *an older **man in a tuxedo** standing next to a younger **man in a tuxedo** wearing glasses.*
- *two men wearing tuxedos glance at each other.*
- *older **man in tuxedo** sitting next to another younger **man in tuxedo**.*

Low Vendi Score, high n-gram diversity:

- *a man and woman cutting a slice of cake by trees.*
- *a couple of people standing cutting a cake.*
- ***the dork with the earring stands next to the asian beauty who is way out of his league.***
- *a newly married couple cutting a cake in a park.*
- *a bride and groom are cutting a cake as they smile.*

Figure 8: Two sets of captions that receive different ranks according Vendi Score and n-gram diversity. We manually highlight some features contributing to the different scores. On the left, a sentence contains repeated n-grams, which are penalized by n-gram diversity. On the right, one long outlier sentence contributes most of the n-grams for this group, greatly increasing the n-gram diversity.

molecules. We evaluate VS using two similarity functions: the Morgan fingerprint similarity (radius 2), and the similarity between odor descriptors, defined as the cosine similarity between descriptor indicator vectors $\phi(x)$, where $\phi_i(x)$ is equal to one if descriptor $i$ is associated with molecule $x$ and zero otherwise.

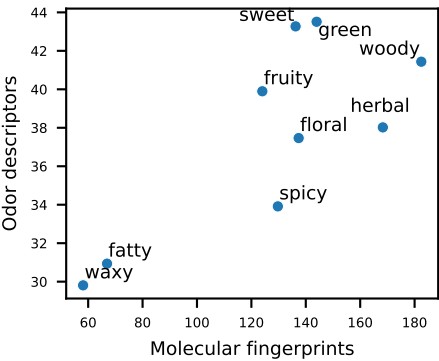

Figure 9: The Vendi Scores of samples containing 500 molecules with different scent labels, calculating diversity using two similarity functions: Morgan molecular fingerprint similarity, and the similarity between odor descriptors. Each molecule is associated with one or more human-written tags (e.g. "floral, fruity, green, sweet"), and the odor-descriptor similarity is the cosine similarity between binary tag indicator vectors.

The diversity scores are plotted in Figure 9. The molecular diversity score and the odor-descriptor diversity scores are correlated, meaning that words like "woody" and "green" are used to describe molecules that vary in molecular structure and also elicit diverse odor descriptions, while words like "waxy" and "fatty" are used for molecules that are similar to each other and elicit similar odor descriptions. For example, the word "green" appears in tag sets such as "aldehydic, citrus, cortex, green, herbal, tart" and "floral, green, terpenic, tropical, vegetable, woody", whereas the word "waxy" tends to co-occur with the same tags ("fresh, waxy"; "fresh, green, melon rind, mushroom, tropical, waxy"; "fruity, green, musty, waxy"). Molecules from the categories with the highest and lowest scores are illustrated in Figure 10.

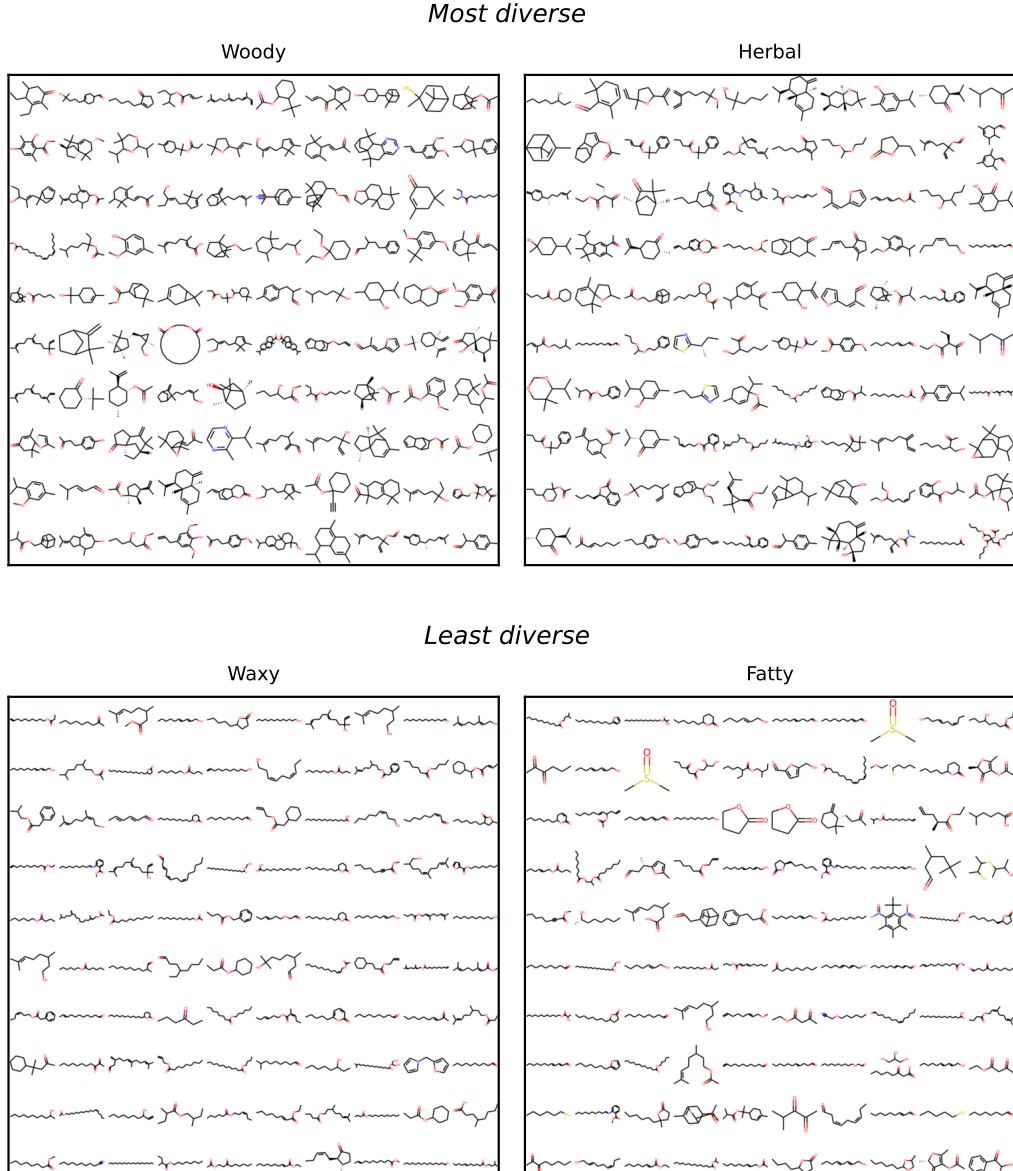

Figure 10: The scent categories in Goodscents dataset with the lowest (top) and highest (highest) Vendi Scores, using the molecular fingerprint similarity. We show 100 examples from each category, in decreasing order of average similarity, with the image at the top left having the highest average similarity scores.

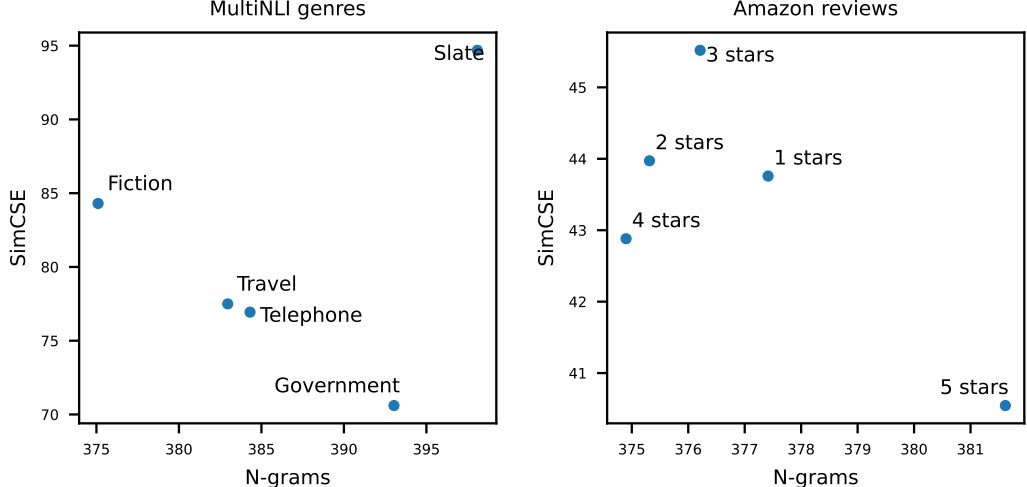

Figure 11: The Vendi Scores of samples containing 500 MultiNLI sentences with different genres (left) or Amazon reviews with different star ratings (right), defining similarity using either n-gram overlap or SimCSE (Gao et al., 2021).

**Text** In Figure 11, we evaluate the diversity scores of samples sentences with different genres, from the MultiNLI dataset (Williams et al., 2018), and Amazon product reviews with different star ratings (Keung et al., 2020), using either the n-gram overlap similarity or SimCSE (Gao et al., 2021). SimCSE is a Transformer-based sentence encoder that achieves state-of-the-art scores on semantic similarity benchmarks. The model we use initialized from the uncased BERT-base model (Devlin et al., 2019) and trained with a contrastive learning objective to assign high similarity scores to pairs of MultiNLI sentences that have a logical entailment relationship.

In MultiNLI, both models assign the highest score to Slate, which consists of sentences from articles published on slate.com. SimCSE assigns a higher score to the "Fiction" category, possibly because it is less sensitive to common n-grams (e.g. "he said"), that appear in many sentences in this genre and contribute to the low N-gram diversity score. In the Amazon review dataset, the 5-star reviews have the highest N-gram diversity but the lowest SimCSE diversity, perhaps because SimCSE assigns high similarity scores to sentences that have the same strong sentiment. SimCSE assigns the highest diversity score to 3-star reviews, which can vary in sentiment.

**Images** Following the setting in Section 4.6, we evaluate two additional datasets, Fashion MNIST (Xiao et al., 2017) and CelebA (Liu et al., 2015). We use the same similarity scores as in Section 4.6. Images in CelebA are associated with 40-dimensional binary attribute vectors. We use these attributes as an additional similarity score, defining the attribute similarity as the cosine similarity between attribute vectors. These illustrations highlight the importance of the choice of similarity function in defining a diversity metric.

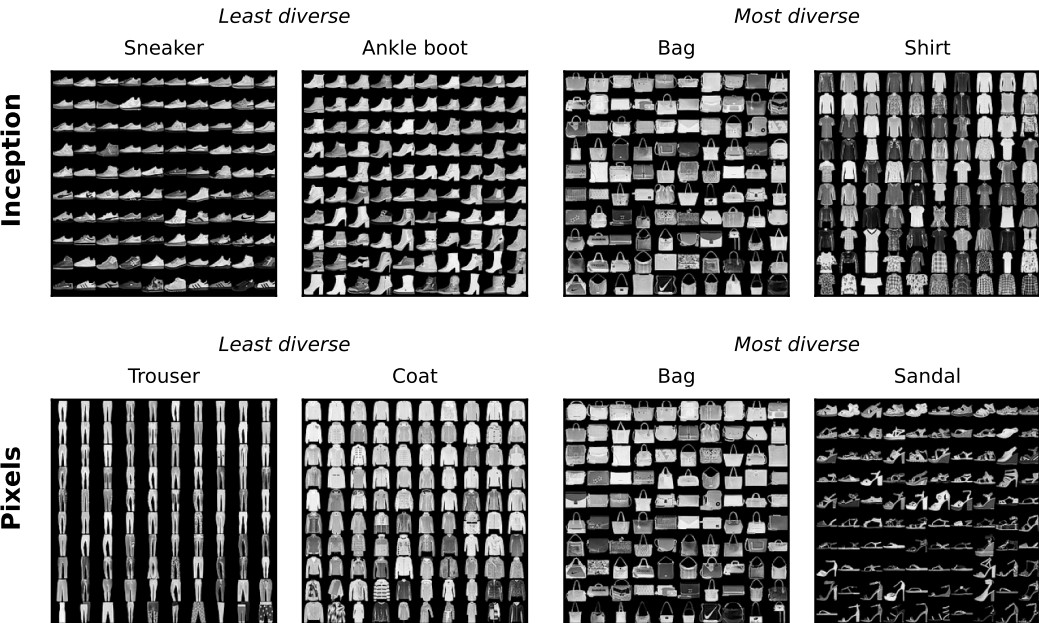

Figure 12: The categories in Fashion MNIST with the lowest (left) and highest (right) Vendi Scores, defining similarity as the cosine similarity between either Inception embeddings (top) or pixel vectors (bottom). We show 100 examples from each category, in decreasing order of average similarity, with the image at the top left having the highest average similarity scores according to the corresponding kernel.

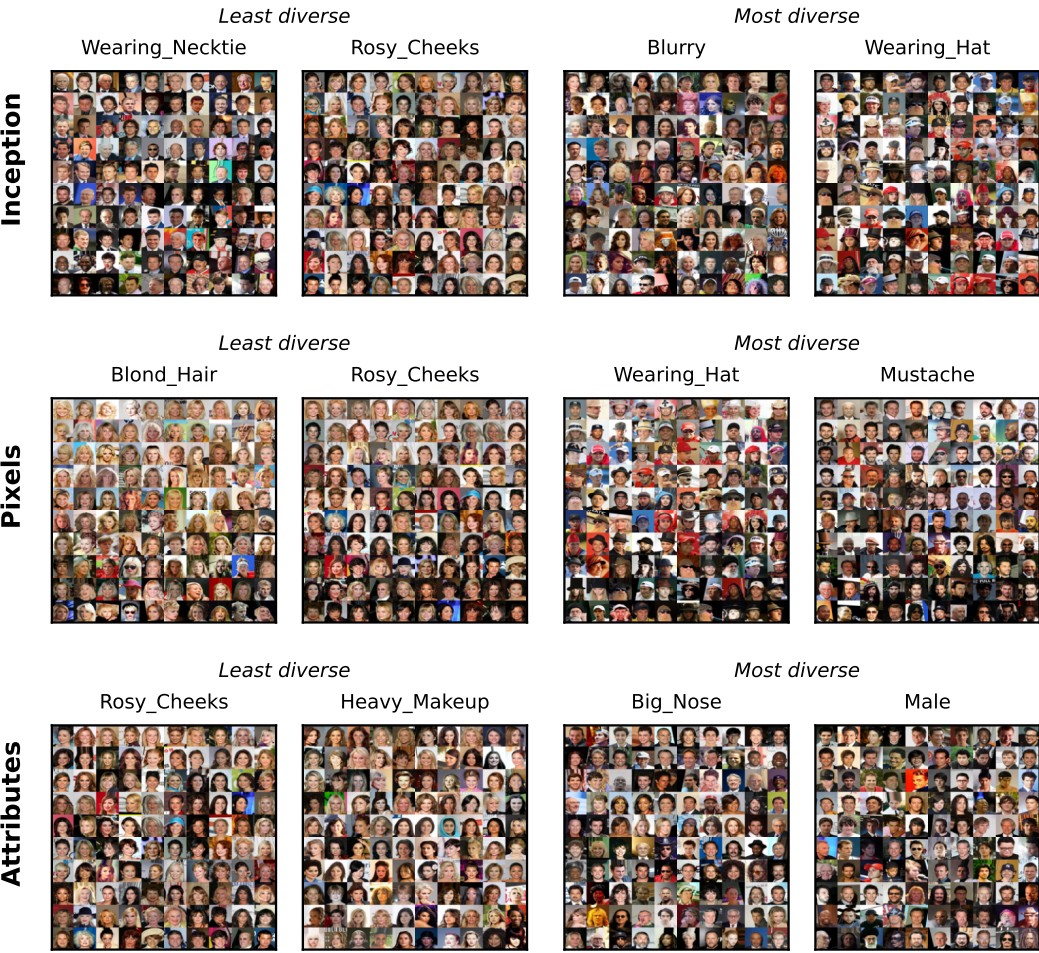

Figure 13: The attributes in CelebA with the lowest (left) and highest (right) Vendi Scores, defining similarity as the cosine similarity between either Inception embeddings (top), pixel vectors (middle), or binary attribute vectors (bottom). We show 100 examples from each category, in decreasing order of average similarity, with the image at the top left having the highest average similarity scores according to the corresponding kernel. These examples illustrate the importance of the choice of similarity score for defining the notion of diversity that is relevant for a given application.

