# OpenReview forum: "The Vendi Score: A Diversity Evaluation Metric for Machine Learning"
_ICLR.cc/2023/Conference — Submitted to ICLR 2023_

### Official Review · Reviewer_Z1XS · 2022-10-18

**Confidence:** 5
**Correctness:** 3
**Technical Novelty And Significance:** 3
**Empirical Novelty And Significance:** 2
**Recommendation:** 5

**Clarity, Quality, Novelty And Reproducibility:**

### Clarity

- C1. The first paragraph of the Introduction could be elaborated, including a definition of diversity, justification of this work, etc.

- C2. The related work in the Introduction section deserves a separate section.

- C3. Why do the authors call it the *Vendi* Score?

- C4. The definition of IntDiv is introduced too later (Sec. 3), although it is referred to in the previous sections frequently.


### Quality

- Formal writing does not abbreviate "does not" to "doesn't." e.g., "the Vendi Score doesn't" in Abstract.

- Appendix could start with alphabetic numbering, not 5.

- Figures and tables are misplaced on the page where it is referred. Although it cannot be perfectly matched, you could try.

### Novelty

By seeing the distribution of eigenvalues as a probability distribution, the authors proposed to use the entropy of these eigenvalues to assess feature diversity was quite interesting. If the experimental validation shows a significant outperformance over conventional competing metrics, it would be a strong paper for the related communities.

**Strength And Weaknesses:**

### Strength

The proposed idea has a compelling theoretical motivation, and it is convenient to use since it only requires the computation of the eigenvalue decomposition for the feature covariance matrix. Through a normalization technique, seeing the distribution of eigenvalues as a probability distribution was notably interesting. The connection to spectral analysis may be worth exploring and discussing.

### Weakness

**W1. Need for human evaluation.** The major concern is experimental validation. Table 1 does not provide the ground-truth diversity; we only learn that VS may capture the HMM's underperformance. Table 2 indirectly validates the VS by showing the sensitivity toward mode detection, not directly on the degree of diversity. Table 3 only compares with the other metrics, which may have some sense of measuring diversity (e.g., FID through its covariance matrices). It just shows general agreements with the other metrics. We do not know which metric is significantly better than one other for measuring diversity. The same goes for Table 4. Even human-written reference captions do not directly confirm that their diversity is captured by BLEU-4, excluding other factors. One suggestion is to evaluate with human judgments on diversity to compare with other conventional and competing metrics.

**W2. Requirement of reference datasets due to inconsistency across domains.** Although the authors argue that it "doesn't require a reference dataset" in the Abstract, it shows inconsistent results across domains in Table 3. For instance, Cat or Bedroom datasets have relatively low diversity compared to that of ImageNet. Therefore, we need the diversity score of the corresponding reference (test) dataset to compare the scores.

**W3. Insensitivity of diversity for small domains.** The discriminative power of the Vendi score seems to attenuate for small domains, e.g., Cat and Bedroom results in Table 3. Maybe, if we finetune the feature extractor (inception networks) on the target domain, one may get better results, but it is not explored.

**W4. A problem in inner products as similarity function.** In Sec. 2.3, you should use *cosine similarity* instead of *inner-product* since the range of the output of the inner-product is not bounded to have ones of diagonal elements for the similarity matrix. Without the l2-normalization of features, the critical assumption of this work goes wrong. Could you confirm that the features are appropriately normalized in all your experiments? You did not explicitly mention it. Also, the covariance matrix is defined by $\mathrm{X}\mathrm{X}^\intercal /n$ only when the features are unbiased (subtracted with their mean vector).

**W5. Connection to the quantum statistical mechanics.** Although the authors mention the connection to quantum statistical mechanics *briefly* in 2.1, it seems to oversell the content of the paper in the Abstract unless additional elaborations or authors' speculations.

**Summary Of The Paper:**

This paper proposed the Vendi Score (VS) to measure the diversity of the generated samples from generative models. It is defined by the exponential of the Shannon entropy of the eigenvalues of a similarity matrix or covariance matrix using the inner-product similarity function (Sec. 2.3).

This work is reminiscent of spectral clustering, where the eigenvalues of the similarity matrix are used for clustering. Here, many clustering means diverse samples. The author can relate this work to spectral clustering and discuss it.

**Summary Of The Review:**

Generally speaking, the proposed idea is appealing and plausible as a general metric for assessing the diversity of generated samples. However, this work could be significantly improved in writing (see the clarity) and experimental validation (see the weakness, especially W1).

---

> ### Author Response · Authors · 2022-11-09
> **Response to Reviewer Z1XS**
>
> Thank you for the helpful comments! We have incorporated your suggestions into our updated draft (please see our Overall Response above).
>
> ### Connection to spectral clustering
>
> Thank you for pointing this out! We have updated the draft with a discussion of the connection to spectral clustering. In particular, the Vendi Score can be related to the number of connected components in a graph.
>
> ### Need for human evaluation
>
> Our focus in this paper was in illustrating the usage of the Vendi score for a variety of ML applications, and we agree that, in these settings, it is difficult to know the ground truth diversity. But please note that we have conducted controlled experiments that validate that the Vendi score captures intuitive notions of diversity in settings where we do have some ground truth knowledge of the number of modes--please see our comment in the Overall Response. Conducting a human evaluation is an interesting direction, but somewhat outside of the scope of this paper, and we leave it to future work.
>
> ### Requirement of reference datasets due to inconsistency across domains.
>
> When we say that VS doesn’t require a reference dataset, we are primarily comparing it to metrics such as FID or Precision/Recall, that define diversity in terms of coverage of a reference dataset. These metrics can only be calculated using both a source and reference dataset. The Vendi Score is defined in terms of a single dataset. It’s true that the scores have different ranges when we compare generative models trained on different datasets, but we don’t actually need to know the diversity score of the test dataset in order to compare the diversity scores between two models.
>
> ### Insensitivity of diversity for small domains.
>
> The fact that the Vendi Scores are lower on some datasets than others (e.g. LSUN Cat vs. ImageNet) is a consequence of the similarity function, not the Vendi Score itself. For example, consider an embedding function that mapped all cat species to orthogonal embeddings, and mapped all non-cat images to a single point. The resulting similarity function would assign high scores to LSUN Cat models and relatively lower scores to ImageNet. This underscores the fact that, like with any similarity-based diversity measure, the values of the Vendi Score depend on the suitable choice of similarity function–which we can see as an opportunity to incorporate domain knowledge into the evaluation metric.
>
> We touch on this point in Appendix Table 5, which compares the similarity scores using the cosine similarity between Inception embeddings and pixel vectors. The pixel diversity scores are on a lower scale, indicating that this similarity metric is less capable of making fine-grained distinctions between the images in these samples.
>
> ### A problem in inner products as similarity function.
>
> Thank you for pointing this out. We do use the cosine similarity in our experiments, along with other similarity metrics (like the probability product kernel) for which self-similarity is always equal to 1. This is stated in the experiment descriptions in section 3, and we have updated the draft to state it more clearly in the method section as well. Regarding the covariance matrix: we do not center the matrix and have updated section 2.3 to reflect that.
>
> ### Connection to quantum statistical mechanics
>
> We have updated the abstract to place less emphasis on the connection to quantum statistical mechanics.
>
> ### Clarity and quality
>
> Thank you for the helpful suggestions. Regarding the name, we chose the name “Vendi Score” because the metric can be seen as the “von Neumann diversity”: defining diversity as the exponential of entropy, VS is the diversity associated with the von Neumann entropy. We have added an explanation to the final draft.
>
> We have also updated the draft to incorporate your other comments: expanding the introduction, separating the related work section, defining IntDiv earlier, avoiding contractions, using letter section titles in the appendix, and matching figures and tables to where they are mentioned in the text.

---

### Official Review · Reviewer_zAvK · 2022-10-23

**Confidence:** 5
**Correctness:** 3
**Technical Novelty And Significance:** 3
**Empirical Novelty And Significance:** 2
**Recommendation:** 5

**Clarity, Quality, Novelty And Reproducibility:**


I believe it is the first time to apply classical bio-diversity scores (specifically, the Hill index of order 1) to several applications in machine learning. However, this work ignores a rich class of diversity related research in machine learning.


**Strength And Weaknesses:**


### Strengths:

- The proposed Vendi score serves as an interpretable metric for diversity.  The authors study several interesting properties of it, which gives a formal understanding of desiderata for diversity.
- The Vendi score enjoys flexibility and wide applicability.
- The Vendi score has the potential to be applied for diversity-informed data augmentation.


### Weaknesses:
- Comparison with  classical diversity tools, e.g.., Determintal point processes （DPPs） is lacked. There is a rich literature on it, stemming from statistical physics and various applications in machine learning.  To name a few:  see [Li et al. 2016] and a wonderful survey [Kulesza-Taskar 2012] and the references therein.

- To make it clearer, if one follow the DPP definition of a diversity score, let us call it the DPP score (DS for short)  following the same notation in the paper, it shall be defined as
$$ DS_k(x_1,…,x_n) = \log |K| = \sum_i \log n\lambda_i = \sum_i (\log n + \log\lambda_i) $$
It looks pretty related to the Vendi score.

Meanwhile, DPP score also has an interpretation, it corresponds to the square of the volume spanned by the vectors of $ x_1,…,x_n$.

Can you comment more on the connections to DPP score?

### Reference:

Alex Kulesza, Ben Taskar, et al. Determinantal point processes for machine learning. Foundations and Trends® in Machine Learning, 5(2–3):123–286, 2012

Chengtao Li, Stefanie Jegelka, and Suvrit Sra. Efficient sampling for k-determinantal point processes. In Artificial Intelligence and Statistics, pp. 1328–1337. PMLR, 2016.



**Summary Of The Paper:**

This paper propose a diversity metric, called Vendi score, that connects and extends ideas from ecology and quantum statistical mechanics to machine learning.

The Vendi Score doesn’t require a reference dataset or distribution over samples or labels, it is therefore general and applicable to any generative model, decoding algorithm, and dataset from any domain where similarity can be defined. The authors demonstrate the usage of Vendi score on domains including molecule generation, mode collapse of GANs, and decoding algorithms for texts.


**Summary Of The Review:**


I appreciate the authors’ efforts in presenting a notion of diversity score motivated by bio-diversity and applying it to various diversity-related applications in machine learning. However, besides the above weakness concerns, I have the following questions for the authors. I believe these questions need to be well-addressed before the work becomes a solid one.

### Q: Considering applications of DPP-based diversity, does Vendi score allow for efficient sampling, marginalization or learning, which are frequently needed for applications in ML?

Specifically, this would be possible if you define an energy-based model based on Vendi score, similar as what DPP does. This energy-based model describes the odds that a specific subset $S$ shall be selected out of the ground set $V$.

$$p(S) \propto VS_k(S) =  \exp{-\sum_i \lambda_i \log\lambda_i},$$

where $S:= \{x_1,…,x_{|S|}  \},  V:={x_1,…,x_{|V|} }, $  (should be the notion of a set, but it does not show up correctly in this openreview system...)

In this way, the partition function (normalization constant) shall be written as:
$$ Z = \sum_{S\subseteq V} VS_k(S) $$

One clear advantage of DPP is that the partition function of it admits a closed form, which is not possible for many other Markov random field variants. Would the distribution defined by the Vendi score allow for similar kind of benefits?


### Q: Would it be possible to axiomatize the diversity metrics?

There could be more than one way to define a diversity score (or even metric mathematically), as we have discussed above.  It is beneficial to think of some axioms that shall be met for a reasonable diversity measure: $D(A)$, where $A = \{x_1, …, x_n\}$ denotes the set of samples.
For example, the following axioms would come to my mind:

1) monotonicity:  $D(A) \leq D(A\cup {x})$
2) symmetry as you defined
3) some relationship amongst $D(A)$, $D(B)$, $D(A\cup B)$, $D(A\cap B)$?
4) defining $D(\emptyset)$?


### Q:  How to correctly evaluate the validity of a diversity score empirically?

The authors did extensive experiments on the usage of Vendi score in various application domains. However, it hardly provides strong evidence of the advantage of Vendi score.   Molecule generation experiments (sec. 3.1) shows one specific instance where IntDiv and VS have discrepancy. While IntDiv is a pretty weak baseline which results from a heuristic definition of diversity.   Image generation experiments (sec. 3.3) shows that VS generally agrees with existing metrics on low-resolution images. And on LSUN dataset, VS disagrees with IS.  Experiments on decoding text (sec. 3.4) show that VS agrees with N-gram diversity on a caption generation task with 5 sentences.  Experiments on diagnosing datasets (sec. 3.5) demonstrate that VS could provide diversity scores for samples within a specific class in several public datasets.

One observation is that there is no, or it is very hard to get “golden” ground truth for diversity scores of a set of samples.  In my opinion, one way could be to ask a group of persons to provide annotations of diversity, either in the form of “scores” or rankings. One practical way to conduct this is to use some crowdsourcing platform.   A second way would be to use existing datasets in a smart way, say the CIFAR-100 dataset. One can make some groups of data samples out of the CIFAR-100, some groups only contain samples from one unique class (maybe called intra-class groups), and some groups have samples from various classes (called inter-class groups). One would naturally expect that the diversity of inter-class groups ranks higher than that of intra-class groups.  One can of course design more refined data groups along this line, to provide more refined way to verify the validity of a diversity score.



### Q: Naming it as “Vendi score” seems to be not consistent with the literature.

On the first glance I thought Vendi might be taken from some literature of bio-diversity. However, it turns out to be no after I checked representative papers on bio-diversity since the term “Vendi” has never appeared in the literature.

The most related concept may be the “Hill number of order 1” (https://en.wikipedia.org/wiki/Diversity_index#cite_note-Tuomisto2010a-4), see also the paper of Hill [Hill 1973]. So it seems to be reasonable to name it as “Hill score”.

### Reference:

Hill, Mark O. "Diversity and evenness: a unifying notation and its consequences." Ecology 54, no. 2 (1973): 427-432.

---

> ### Author Response · Authors · 2022-11-09
> **Response to Reviewer zAvK**
>
> Thank you for the thorough review and the helpful comments! We have incorporated your suggestions into our updated draft (please see our Overall Response above).
>
> ### Comparison to DPPs
>
> Thank you for the suggestion to discuss DPPs. DPPs are indeed relevant to the Vendi score and we have updated the draft to include the following discussion (Section 2). While DPPs have been widely studied for diverse subset selection, to our knowledge, the DPP score has not been proposed for *evaluating* diversity. For instance, to cite one recent example in NLP, Meister et al. (2021) use DPPs to construct a diverse sampling algorithm, but evaluate the results using the standard n-gram overlap method. As an evaluation metric, the DPP score has some limitations:
> * The DPP score is undefined when the similarity matrix is singular (i.e., if the sample contains any duplicates).
> * Interpretation: While the DPP score has an interpretation related to the volume of the feature space, we would argue that this meaning is not so easy to interpret (for example, what are the units?). On the other hand, the Vendi score can be understood in terms of the effective number of unique elements in a sample.
> * Connection to entropy: We choose to follow an approach developed in biodiversity and define diversity in terms of entropy, which is a property of a distribution. This is the approach that makes most sense for the applications we consider, namely evaluating generative models. The DPP score is a property of an event (the likelihood of drawing subset $X$ from reference set $\mathcal{X}$).
>
> [1] Clara Meister, Martina Forster, Ryan Cotterell (2021). Determinantal Beam Search.
>
> ### Efficient sampling, marginalization or learning
> Our focus is on evaluating the diversity of a collection of elements, rather than sampling or optimization. Sampling and marginalization are not clearly relevant in our context, as we are primarily interested in measuring the diversity of distributions (rather than selecting subsets of distributions). The Vendi Score can be incorporated into learning algorithms – for example, in gradient based optimization – but we believe this is outside the scope of this paper and plan to investigate this direction in future work.
>
> ### Would it be possible to axiomatize the diversity metrics?
> In section 2.2, we discuss some properties that we believe are desirable for a diversity metric to have, including symmetry, and a property relating to the relationship between $D(A), D(B), D(A \cup B)$ (the partitioning property). These constitute our axioms for a diversity measure.
>
> On the other hand, the Vendi score does not satisfy the monotonicity principle, and we would argue that diversity is not necessarily monotonic. This argument is made in the context of biodiversity by Leinster and Meckes (2016) (whom we will cite in the updated draft): Consider a forest containing one species of oak and nine species of pine. Introducing another species of pine to this forest will decrease the diversity score, because it means that the forest is more heavily dominated by pines.
>
> [1] Leinster and Meckes, 2016. Maximizing Diversity in Biology and Beyond
>
> ### How to correctly evaluate the validity of a diversity score empirically?
> Our focus in this paper was in illustrating the usage of the Vendi score for a variety of ML applications, and we agree that, in these settings, it is difficult to know the ground truth diversity. But please note that we have conducted controlled experiments like the one you describe to validate the Vendi score in settings where we do have some ground truth knowledge of the number of modes. These experiments appear in the appendix, but have updated the draft to highlight them more prominently in the main paper (please see our Overall Response above). In particular:
> * Synthetic data: In Figure 2 (Appendix Figure 5 in the original submission), we compare the Vendi Score with IntDiv on univariate mixture-of-normal distributions, varying either the number of components, the mixture proportions, or the per-component variance, and show that VS behaves consistently and intuitively in all three settings.
> * Realistic data: In Appendix Figure 5 (Appendix Figure 6 in the original submission), we create groups of datasets out of two classification datasets (MNIST and MultiNLI), with each group containing examples from an increasing number of classes. We verify that VS increases with the number of classes.
>
> ### Naming
>
> The Vendi Score is indeed related to the Hill number of order 1. However, the Hill numbers are functions of categorical probability distributions, while our metric is a function of a similarity matrix, so it may be misleading to reuse that name here. We chose the name “Vendi Score” because the metric can be seen as the “von Neumann diversity”: defining diversity as the exponential of entropy, VS is the diversity associated with the von Neumann entropy. We have added an explanation to the final draft.

---

### Official Review · Reviewer_7smz · 2022-10-25

**Confidence:** 3
**Clarity, Quality, Novelty And Reproducibility:** The paper is easy to read and well-wr…
**Correctness:** 4
**Technical Novelty And Significance:** 2
**Empirical Novelty And Significance:** 2
**Recommendation:** 5

**Strength And Weaknesses:**

Strength:

1. Paper shows many applications of the Vendi score to measure the diversity of data set, i.e. in generative model, mode collapse, image evaluation, algorithm evaluation etc.
2. The Vendi score requires no label to compute.

Weaknesses:

1. The definition of Vendi score comes straight forward from the Shannon entropy. All of its theoretical results are consequences of results of Shannon entropy.

2. There is no theoretical support for the use of Vendi score in situations presented in the paper.

3. There is no guideline to explain when to use the Vendi score, i.e in the case of LSUN dataset, should we use the Vendi score to rank the efficiency of models?


To improve the work, the authors could prove that the Vendi score is an indicator for the number of modes.


**Summary Of The Paper:**

The paper proposes to use Vendi score to measure the diversity of datasets.

**Summary Of The Review:**

I am concerned about the novelty of the paper, since most of the contributions come from experimental results.

---

> ### Author Response · Authors · 2022-11-09
> **Response to Reviewer 7smz**
>
> Thank you for the helpful comments.
>
> > The definition of Vendi score comes straight forward from the Shannon entropy. All of its theoretical results are consequences of results of Shannon entropy.
>
> The definition of the Vendi score is closely related to the Shannon entropy, but we do not see this as a weakness of the proposed metric. On the contrary, the connections to entropy provide tools for reasoning about the properties of the metric and connecting it to existing work in biodiversity. We would also emphasize that the Vendi score is related to the spectral entropy of a similarity matrix; this similarity-sensitive notion of entropy has important differences from the Shannon entropy. Most of all, this metric has not previously been explored for measuring diversity in machine learning.
>
> > There is no theoretical support for the use of Vendi score in situations presented in the paper… To improve the work, the authors could prove that the Vendi score is an indicator for the number of modes.
>
> We prove that the Vendi score is equal to the effective number of completely dissimilar elements in a set, which we argue (following biodiversity) is a reasonable definition of diversity, and which provides a justification for the use of this metric.
>
> Moreover, we also prove a partitioning property that specifies the relationship between VS and the number of modes in the case that the modes are well-separated. Specifically, the partitioning property in theorem 2.1 states: suppose that the sample $S$ can be partitioned into m subsets $S_1, \ldots, S_m$ with relative sizes $p_1, \ldots, p_m$, such that for any $i \neq j$, for all $x \in S_i, x’ \in S_j$, $k(x,x′) = 0$. Then VS of the combined sample is given by the geometric mean, $VS_k(S_1, \ldots, S_m) = \exp(H(p_1, \ldots, p_m)) \prod_{i=1}^m VS_k(S_i)^{p_i}$. This corresponds to the case where the dataset consists of $m$ dissimilar modes. If we consider the simple case where all modes have equal size, and all elements within each mode are identical to each other, then VS is exactly equal to $m$, the number of modes. (VS increases if there is greater within-mode diversity.) These properties provide theoretical justification for our use of VS to measure diversity. They are illustrated with a toy example in Figure 1.
>
> Please note that we also empirically illustrate the relationship between VS and the number of modes in the data in Figure 2, and Appendix Figure 5 in our updated draft. Please see our Overall Response for a discussion of these experiments.
>
> > There is no guideline to explain when to use the Vendi score, i.e in the case of LSUN dataset, should we use the Vendi score to rank the efficiency of models?
>
> We chose a varied number of experiments to showcase use cases for the metric, e.g. to measure mode collapse, to compare the diversity of different models and datasets, and we would argue that yes, we can use the Vendi score to rank the diversity of models on the LSUN datasets. Can you please clarify what other kinds of guidelines you would suggest we provide?

---

### Author Response · Authors · 2022-11-09
**Overall Response**

We thank the reviewers for their thorough comments and suggestions. We are glad that all reviewers appreciate the flexibility and applicability of the proposed metric. The reviewers made a number of great suggestions. We have incorporated these into our draft and uploaded an updated version of the paper.

In particular, Reviewer zAvK and Reviewer Z1XS raised concerns about whether our experiments validate the use of the Vendi Score, given that we can’t always have a ground-truth measurement of diversity as a reference. We would like to note that we have conducted controlled experiments in settings where we do have some ground truth knowledge of the number of modes:
* Synthetic data: In Figure 2 (Appendix Figure 5 in the original submission), we compare the Vendi Score with IntDiv on univariate mixture-of-normal distributions, varying either the number of components, the mixture proportions, or the per-component variance, and show that VS behaves consistently and intuitively in all three settings.
* Realistic data: In Appendix Figure 5 (Appendix Figure 6 in the original submission), we create groups of datasets out of two classification datasets (MNIST and MultiNLI), with each group containing examples from an increasing number of classes. We verify that VS increases with the number of classes.

These experiments appeared in the appendix in our original submission due to space constraints (we wanted to showcase the metric in a wide range of scenarios), but we have updated our draft to highlight them more prominently in the main paper. We would also point to Figure 1, which illustrates how VS changes as a function of the similarity matrix in a toy setting where we can specify the similarity function precisely.

We have also revised the introduction to address some clarity issues identified by Reviewer Z1XS, creating a separate related work section, and we have expanded the related work to discuss the topics suggested by the reviewers (determinantal point processes and spectral clustering). We respond to the reviewers individually below.

---

### Author Response · Authors · 2022-11-23
**Gentle reminder about the discussion**

Dear reviewers, thank you again for your constructive feedback. We were wondering if you have considered our rebuttal and whether you have any questions for us. Thank you.

---

### Decision · Program_Chairs · 2023-01-20

**Decision:**

Reject

**Justification For Why Not Higher Score:**

There was a consensus among reviewers that this paper should be rejected.

**Justification For Why Not Lower Score:**

N/A

**Metareview: Summary, Strengths And Weaknesses:**

This paper proposes to use the Shannon entropy of the eigenvalues of the feature similarity matrix to evaluate diversity of samples of a generative model. While the idea was deemed interesting, there was a consensus among reviewers that this paper should be rejected. The key concern is how useful the score is, in particular since the utility of the score has not been assessed using human evaluations. Furthermore, the value of the score can be different across data sets, making it hard to know reasonable values without having access to a ground-truth test set.